# Shared Decision Making with Acutely Hospitalized, Older Poly-Medicated Patients: A Mixed-Methods Study in an Emergency Department

**DOI:** 10.3390/ijerph19116429

**Published:** 2022-05-25

**Authors:** Pia Keinicke Fabricius, Anissa Aharaz, Nina Thórný Stefánsdóttir, Morten Baltzer Houlind, Karina Dahl Steffensen, Ove Andersen, Jeanette Wassar Kirk

**Affiliations:** 1Department of Clinical Research, Copenhagen University Hospital—Amager and Hvidovre, 2650 Hvidovre, Denmark; anissen9@gmail.com (A.A.); nina.thorny.stefansdottir@regionh.dk (N.T.S.); morten.baltzer.houlind@regionh.dk (M.B.H.); ove.andersen@regionh.dk (O.A.); jeanette.wassar.kirk@regionh.dk (J.W.K.); 2Department of Clinical Medicine, University of Copenhagen, 2200 Copenhagen, Denmark; 3The Capital Region Pharmacy, 2730 Herlev, Denmark; 4Department of Drug Design and Pharmacology, University of Copenhagen, 2100 Copenhagen, Denmark; 5Center for Shared Decision Making, Lillebaelt University Hospital of Southern Denmark, 7100 Vejle, Denmark; karina.dahl.steffensen@rsyd.dk; 6Institute of Regional Health Research, Faculty of Health Sciences, University of Southern Denmark, 5000 Odense, Denmark; 7Emergency Department, Copenhagen University Hospital—Amager and Hvidovre, 2650 Hvidovre, Denmark; 8Department of Public Health, Nursing, Aarhus University, 8000 Aarhus, Denmark

**Keywords:** shared decision making, older patients, polypharmacy, mixed methods, emergency department

## Abstract

Shared decision making (SDM) about medicine with older poly-medicated patients is vital to improving adherence and preventing medication-related hospital admissions, but it is difficult to achieve in practice. This study’s primary aim was to provide insight into the extent of SDM in medication decisions in the Emergency Department (ED) and to compare how it aligns with older poly-medicated patients’ preferences and needs. We applied a mixed-methods design to investigate SDM in medication decisions from two perspectives: (1) observational measurements with the observing patient involvement (OPTION 5) instrument of healthcare professionals’ SDM behavior in medication decisions and (2) semi-structured interviews with older poly-medicated patients. A convergent parallel analysis was performed. Sixty-five observations and fourteen interviews revealed four overall themes: (1) a low degree of SDM about medication, (2) a variation in the pro-active and non-active patients approach to conversations about medicine, (3) no information on side effects, and (4) a preference for medication reduction. The lack of SDM with older patients in the ED may increase inequality in health. Patients with low health literacy are at risk of safety threats, nonadherence, and preventable re-admissions. Therefore, healthcare professionals should systematically investigate older poly-medicated patients’ preferences and discuss the side effects and the possibility of reducing harmful medicine.

## 1. Introduction

Patient-centered care is a goal in healthcare policy worldwide, considered key to improving patient outcomes [1,2] and a prerequisite for medication adherence and patient safety [3]. Shared decision making (SDM) is a cornerstone in patient-centered care [4] and is defined as a collaborative process in which patients and clinicians make healthcare decisions together, considering the best scientific evidence available and the patient’s values, goals, and preferences [5]. Therefore, SDM is a social, collaborative process [5] that is illustrated in the revised “Three-talk model of SDM” by Elwyn et al. [6]. The model depicts a process of collaboration and deliberation with the basic SDM key steps: ”team talk”, “option talk”, and “decision talk” [6].

Older patients with multimorbidity and polypharmacy (≥five prescribed medications) [7,8] are a growing population and represent a public health challenge both in Denmark and globally [9,10]. These patients have frequent admissions to emergency departments (ED), and many of these acute hospitalizations are suspected to be medication-induced [11]. However, SDM about medicine in the ED may be particularly hard to achieve [12]. Nevertheless, research suggests that SDM may be particularly beneficial for vulnerable patients with complex illness and care needs [13,14], such as older patients with multimorbidity and polypharmacy [15,16]. For vulnerable populations, SDM empowers the patient to better understand treatment options and tailor decisions to their individual preferences [17].

A Danish study found that patients with cancer who are socially vulnerable generally want to be involved in their treatment but to varying degrees. Furthermore, some patients who are vulnerable may have difficulty expressing their wishes and needs [18]. Moreover, another Danish survey with 963 patients with diabetes revealed that 71% of patients with diabetes preferred that the healthcare professional and the patients collaborate about treatment decisions. Twenty percent preferred that the healthcare professional presented the possibilities, thus letting the patient make the decision on his/her own, and only eight percent wanted to let the healthcare professional make the decision alone after listening to the needs of the patient [19].

Practicing SDM with older vulnerable patients with several co-morbidities is more difficult than practicing SDM with patients with one disease [16]. A study by Levinson et al. (2005) found that patients’ preferences for an active role in treatment decisions increased with age up to 45 years but then declined for older patients [20]. Some studies indicate that some older poly-medicated patients prefer not to be involved in decisions about their medicine [20,21]. In addition, predicting or giving advice to a patient with multiple diseases and polypharmacy is complex as polypharmacy increases the risk of adverse drug reactions and interactions [16]. Furthermore, multimorbidity and polypharmacy are more frequent among patients with low social status and low health literacy, making their ability to obtain, process, communicate, and understand basic health issues challenging [22].

In recent years, there has been a growing interest in implementing SDM in the ED [23,24,25] because more than 70% of patients are discharged directly from the ED without further treatment in a specialized ward [26]. Several studies highlight that many patients would like to become involved in decisions about their treatment in the ED and to be made aware of this possibility [27,28]. Nevertheless, SDM in the ED setting also presents unique ethical challenges due to the patient’s acute situation and inherent time pressure complicating medical decision making. We recently published an ethnographic study showing that the ED is a challenging context for patient involvement in medicine due to high flow, a fragmented medication process, time pressure, acute situations, and high complexity around older patients with polypharmacy [29]. Thus, introducing SDM in the context of an ED requires specific consideration of factors influencing patient and healthcare professionals’ communication [30]. 

However, to our knowledge, no prior studies have provided insight into SDM in medication decisions in the ED setting, and we found no studies about older poly-medicated patients’ preferences for engaging in decisions about their medicine in the ED that could guide us in tailoring an SDM patient decision aid to support medication decisions.

## 2. Materials and Methods

### 2.1. Objectives

The overall aim was to provide insight into the extent to which SDM is used in medication decisions in the ED setting and compare how it aligns with older poly-medicated patients’ preferences and needs. The overall aim was clarified by combining findings from two sub-studies, one quantitative and one qualitative, and assessing their parallels and differences.

The aim of the quantitative sub-study was to examine to what degree healthcare professionals’ practice SDM in medication conversations in the medical ED.

The aim of the qualitative sub-study was to explore older poly-medicated patients’ experiences of being involved in decisions about their medicine in the medical ED.

### 2.2. Study Design

In this study, we sought to expand our understanding of the complexities of SDM with older poly-medicated patients in the ED. We investigated SDM in medication decisions from two different perspectives, using a convergent parallel mixed-method design inspired by Creswell and Clark [31] integrating two different methods and analyses: a quantitative perspective of the healthcare professionals’ SDM behavior by measuring observational data, and a qualitative perspective interviewing older poly-medicated patients about their experiences of SDM in medication conversations in the ED. With this, we conducted parallel sampling with different people in the two datasets [31].

The quantitative and qualitative data were collected and analyzed separately and then integrated for coherence and differences by drawing meta-inferences from the qualitative and quantitative findings. 

The convergent parallel design process on data collection, analysis, and interpretation is visualized in Figure 1.

Our study intent was to adhere to best practices for mixed-methods research and was guided by the quality reporting guideline checklist for mixed-methods studies by Fetters and Molina Azorin [32], which is attached in Appendix A.

### 2.3. Setting

The study was conducted at a university hospital in the capital region of Denmark. The observations were conducted from October 2018 to February 2019 and August 2019 to September 2019. The interviews were conducted in January 2020. The healthcare system in Denmark is publicly funded by taxes and provides free access for all citizens requiring medical care in hospitals and home care services. The university hospital is divided into three different locations, each hosting a medical ED to cover the catchment area of 517,000 people. The EDs have an average daily intake of approximal 30–45 patients hospitalized for up to 48 h in the ED.

### 2.4. Ethical Considerations

The Danish Data Protection Agency (file number VD-2019-264) approved this study. According to Danish legislation, research that does not address biological material does not require official approval from the Ethics Committee. The participants gave informed written and oral consent to participate. All participants were guaranteed anonymity and the right to withdraw from the study at any time. Hereby, our work adheres to the World Medical Association’s ethical criteria for medical research involving human participants 

### 2.5. Quantitative Data

To access the extent of SDM practice in medication conversations in the medical ED, an observer instrument was used. The validated OPTION 5 (described below) observer instrument [33] was used in situations in which decisions about the patient’s medicine were made in order to examine how SDM was used in conversations and how healthcare practitioners involve their patients in the decision-making process.

### 2.6. Quantitative Data Collections

#### 2.6.1. Participants

The quantitative data consisted of systematic observations of medication communication exchanges collected as a part of a prior ethnographic baseline study in two medical EDs conducted by P.F. from October 2018 to September 2019. The observations followed James Spradley’s participant approach, locating social situations in which communication about medicine occurred. In total, 31 different multidisciplinary healthcare professionals were observed during daily medicine practice. The participants included ED physicians, geriatric physicians, pharmacists, pharmaconomists, nurses, and medical physician specialists [29]. We planned the recruitment of participants in the ethnographic study with the managing nurse, who helped to locate participants according to the inclusion criteria, which was to include participants (healthcare professionals) from different professions, who expected to have a role in the medication process, and with the most diverse clinical experience.

#### 2.6.2. Observing Patient Involvement Measurement (OPTION 5)

The observations consisted of field notes with detailed descriptions of what was said between a patient and the healthcare practitioners about medicine, word by word. The observations were rated with the “Observing Patient Involvement Measurement”(OPTION 5), which is an observer-based measure of SDM [34]. OPTION 5 is based on five key behaviors/steps (items) that constitute SDM: (1) presenting options, (2) establishing a partnership with the patient, (3) describing pros and cons of options, (4) eliciting patient preferences, and (5) integrating patient preferences into the decision [6]. Each item is rated on a Likert scale from 0–4, with 4 illustrating the highest possible degree of SDM. 

To evaluate the healthcare professionals’ overall degree of SDM, we first scored the SDM process for each of the 65 decision-making situations. This was performed with the original items’ score on a 0–4 scale, which was summed in each of the 65 decision-making situations. Then, the overall scores were re-scaled to an overall OPTION 5 score of 0–100, as recommended in the OPTION 5 manual [34] and previous observer OPTION studies [35]. 

In this study, special attention was also paid to each of the five items, as they could reveal detailed new knowledge of older poly-medicated patient preferences and needs for involvement in decisions about their medicine in an acute setting, which is useful knowledge when tailoring SDM to medication conversations in the ED. Points given within the scale between 0–4 were summed for each item and mean point for each item was calculated.

Before initiation of the study, the OPTION 5 manual was cross-culturally translated from the original validated English version to a Danish version. This procedure followed the recommendations for the translating process described by Elwyn [36] to ensure consistency among versions of the questionnaire and was consistent with the leading translation methods in the survey research field. The Danish translation was forward–backward translated by authors P.F. and N.T.S. and commented and accepted by co-authors O.A. and K.D.S., experienced researchers in quantitative methods. However, the Danish version was not further validated in this study. Therefore, along with the Danish translation, a thorough and systematic assessment of the OPTION 5 manual was conducted, accessing each item until agreement was established on keywords and elements, resulting in a Danish consensus scoring manual, which was developed with supervision from three experienced clinical pharmacists with a research background.

Before scoring the field notes, the two raters’ training was conducted in a small pilot study scoring 10 decisional medication situations and comparing the results. According to the consensus scoring manual, the field notes were scored independently by two raters (authors P.F. and N.T.S.). We defined a decisional medication situation as: “A decision about medicine which could either be medication, which was prescribed, reduced, paused or deprescribed”. The decision could also be about non-pharmaceutical remedies, such as physiotherapy, heating pads, and so forth, the impact of which can be comparable with pharmaceutical treatment and was, therefore, also included. We calculated the average score in case of disagreement between the two raters, when no consensus could be reached. 

### 2.7. Quantitative Analysis and Statistics

A statistician (T.K.) conducted the statistical analysis and used basic descriptive statistics [37], which included frequency, averages (mean or median), and ranges such as the interquartile range. All statistical analysis was performed in R 3.6.1 (R Foundation for Statistical Computing, Vienna, Austria).

### 2.8. Qualitative Data

The qualitative data consisted of semi-structured telephone interviews with older poly-medicated patients, seeking insight into the older poly-medicated patients’ needs and preferences for being involved in decisions about their medicine. The qualitative methods and analysis were partly guided by SRQR guidelines [38].

### 2.9. Qualitative Data Collection

#### 2.9.1. Participants

The inclusion criteria were patients 75 years or older who took at least five medications and were discharged. Exclusion criteria were patients with dementia or patients who did not speak or understand Danish. Author A.S. contacted and recruited the patients in the ED, who gave informed consent to participate. Initially, 22 patients accepted to participate, but five withdrew due to severe health problems, and one patient died before the interview. In addition, two patients did not answer the telephone at the scheduled interview time. Data saturation was reached with a total of 14 patients who were included in the qualitative sub-study.

The interviews were originally planned to be conducted in the patients’ homes immediately after they were discharged. However, due to the COVID-19 situation, several older patients were anxious and preferred to be interviewed over the telephone, which was respected. The characteristics of the patients who accepted to participate in the interview are presented in Table 1.

#### 2.9.2. Semi-Structured Interviews

Authors P.F. and A.S. developed a semi-structured interview guide inspired by Kvale and Brinkmanns [39] containing semi-structured questions investigating how the patient had experienced (or not experienced) being involved in conversations about their medicine in the ED. In addition, the three questions from the Collaborate Questionnaire [40], a patient-reported experience measure of SDM, inspired some of the questions in the interview guide and were modified to open-ended, qualitative questions. The original Collaborate questions were: (1) “*How much effort was made to help you understand your health issues?”,* (2) “*How much effort was made to listen to the things that matters most to you about your health issues?”, and* (*3*) *“How much effort was made to include what matters most to you in choosing what to do next?”.* See the modified questions and interview guide in Table 2.

A small, pilot study using cognitive interviews [41] was conducted with two older patients to test the interview guide for comprehension. The two patients understood that the original term “Health Issue” also consisted of the patient’s medicine and was incorporated implicitly in our qualitative question. However, one of the patients explained that both questions were too long and needed repetition and further explanation to understand the question entirely. This patient feedback was carefully incorporated when conducting the semi-structured interviews. Furthermore, the interview guide was validated with J.W.K., an experienced qualitative researcher. In total, author P.F. conducted 14 interviews between mid-January and mid-March 2021, and each lasted an average of 30 min (20–40 min). All interviews (except the first) were conducted as telephone interviews and were recorded and transcribed verbatim.

### 2.10. Qualitative Analysis

#### Thematic Analysis

The patient interviews were analyzed using Braun and Clarks’ reflexive thematic analysis with an inductive approach from the beginning to search openly for patterns in the data [42,43]. First, authors P.F. and A.S. read and re-read the transcriptions to become familiar with the content. Then, text pieces were coded and re-coded in an ongoing process and frequently discussed with A.S. and co-author J.W.K. Later, the coded material was gathered into initial themes and sub-themes, which were renamed several times. Finally, we identified five themes and 12 sub-themes in the thematic analysis, which are presented in Appendix A. An example from the qualitative coding process is also presented in the Appendix A.

### 2.11. Integration of Data

SDM is a social, collaborative process where two persons participate: a patient and a healthcare professional [5]. In our study, we explored SDM in decisions about medicine in the ED from two perspectives: a quantitative observational perspective of the healthcare professionals’ behavior and a qualitative patient perspective.

The quantitative and qualitative data were merged at the interpretation and reporting levels. The analysis illustrates how well the qualitative and quantitative findings matched or differed.

## 3. Results

### 3.1. Participants

A total of 31 different healthcare professionals were observed in a total of 65 medication decisional situations, and 14 patients participated in the semi-structured interviews. The participants characteristics are presented in Table 3.

### 3.2. Quantitative Results Presenting Option 5 and Items’ Scores

The 65 medication decisional situations and five items were scored as shown in Table 4.

### 3.3. Integrating Quantitative and Qualitative Data

Our result presented four overlapping themes presented in a narrative, weaving coherent text combining selected elements from the quantitative and qualitative data [44] to shed light on the central SDM process with collaboration and deliberation about medication decisions in focus. The original five themes and 12 sub-themes from the qualitative analysis were reduced to four overall themes when merged with the quantitative OPTION 5 results. The quantitative and qualitative results are presented together on a theme-by-theme basis as suggested by Fetters et al. [44]. The meta-inferences between the two datasets are visualized in Table 5.

### 3.4. Overall Themes including Both Quantitative and Qualitative Data

We identified four overall themes, which seem to characterize SDM in medication decisions in the ED setting. The four themes are:A low degree of SDM in conversations about medicine.A variation in the pro-active and non-active patient approach to conversations about medicine.No information on side effects.A preference for medication reduction.

#### 3.4.1. Theme 1: A Low Degree of SDM in Conversations about Medicine in the ED 

The OPTION 5 measure of SDM reveals that healthcare professionals have a low level of SDM in medication conversations in the ED, as presented in Table 6. With an overall low mean score of 8.2 (out of 100 possible), within the range of 0.0–5.0, and a standard deviation range of 16.2, the healthcare professionals have a low level of SDM in decisions about medicine in the ED.

The qualitative interviews reflected that many older patients were unaware that there could be different treatment options, which is revealed in the low average of 0.53 points on Item 1 “Presenting Options”. Nevertheless, Item 1 was the SDM behavior in which the healthcare professionals performed best.

In the interviews, several older patients explained that they had not experienced much communication about medicine in the ED at all. Some patients reflected upon the lack of medication conversations and thought it was due to a lack of time in the ED and their acute health condition, making medicine conversations less prioritized and challenging to remember.

When directly asked if the patient remembered medication conversations in the ED, one patient replied: 

“*No, I don’t think so, but it appeared we should hurry, I’m not sure, I’m not sure. [……….] No, I didn’t think we were talking about medicine, at all, but it is possible I was a little dazed when I first came in” (Pt ID #16)*.

Poor memory of what happened in the ED could influence the patient’s experience of medication conversations in the ED, which became apparent when one of the interviewed patients first stated that she had not received any medication communication in the ED. Yet, 30 min after the interview, the patient texted P.F. that she forgot that she did have a thorough medication review in the ED and she had been very much involved, but she almost forgot about it until she participated in the interview.

Another patient rationalized the absence of medication communication by claiming that there was no need for medication dialogues because physicians could read about the patients’ medications on the computer. This is illustrated in the following patient quote.

*Interviewer: “When you were in the Emergency Department, did anyone discuss your medication with you?” Patient: “No, they have everything in writing, so there is nothing to discuss. They simply glance at their screens to know what you get and don’t get” (Pt ID #2)*.

However, the interviews also revealed that several patients explained that they had confidence in their medicine and, hence, they did not actively question it.

#### 3.4.2. Theme 2: A Variation in the Pro-Active and Non-Active Patients’ Approach to Conversations about Medicine

Item 2: “Establishing a partnership with the patient” was low, with a mean point of 0.17, indicating that the healthcare professionals in the ED do not systematically invite the patients to collaborate on medication decision making. However, the interviews revealed variations in the patients’ behavior, how they acted, and how they perceived the lack of medicine conversations in the ED. There was a variation in how much initiative the patients took themselves to obtain the information and involvement that they needed. Some patients had much initiative, those we name “pro-active” patients, who said they were very involved, while other patients (the non-active) did not raise any questions or initiative to enter a dialogue about their medicine in contrast to the pro-active patients. Moreover, the non-active patients were divided into two groups: patients who preferred to leave medical decisions to the healthcare professionals and patients who were afraid of revealing that they were not in control of their medicine.

The following is a quote from an active, high-health-literacy patient, a retired physician. When asked if he had been involved in decisions about medicine in the ED, this pro-active patient responded: 

“*Yes, indeed, I believe I was engaged because they listened and heard what I had to say. I communicated my dissatisfaction and worry with the situation and my attitude toward it (the medicine). To me, it’s natural” (Pt ID #21).*

This high-health-literacy patient had experienced that the healthcare professionals had listened to him carefully and elicited his preferences to a great extent, which was in contrast to what the majority of patients in the ED had experienced.

In contrast, several patients who had a non-active approach stated in the interviews that they just followed the advice of the healthcare professionals regarding their medicine without questioning it because they thought that they did not have sufficient knowledge about medicine to enter into a discussion. This is illustrated in the following quote from a non-active patient. 

“*I’ve only just picked up on what was said to me. That is something I have to admit. I presume that the physicians who have dealt with me in the situations, that they are most familiar with what is the best. So, with what is required, I replied: “Yes, thank you.” But because I have no sense of medicine, I hear what the experts say to me, right?” (Pt ID #19)*.

Two non-active patients stated that patient involvement in medicine could signal that they were not controlling their medication, which they seemed to hide. This is illustrated in the following patient quote; when asked about her attitude to being involved in her medicine, the non-active patient responded: 

“*Well, it (involvement) would be wonderful for someone who does not have much control over it and is, if I may say so, gullible” (Pt ID #17).*

Another non-active patient also expressed why she would not like to become involved in her medicine. This is reflected in the following quote.

“*No, I don’t want to get involved, but the day when I am unable to care for myself, I would like to be involved in my medication” (Pt ID #5)*.

The consequence of lack of medicine communication between older patients with a non-active approach and their healthcare professionals is illustrated in the situation of one patient. The patient first became aware that he had not been informed about the effect of a new medication at the time of the interview 1 week after discharge from the ED. 

This is illustrated in the following quote from a non-active patient. 

“*I just got home, and my medicine says right here (on the medicine list) Losartan, which is the blood pressure medication I take once a day. The patient examines the medicine list attentively and states: It does not say whether it is for the blood pressure to rise or fall? I’m not sure of that” (Pt ID #11)*.

The low Item 3 mean score (discussing pros and cons of options) of 0.37 also supports our findings, that many patients in the ED were not informed about the pros and cons of their new medicine before discharge from the ED. This places non-active patients, who do not ask questions by themselves, at high risk of non-adherence and medication errors, when the physicians do not check if the patient does understand the given information about the new medicines.

On the contrary, some pro-active patients were fully aware when they did not receive the needed information on new medicine. This could drive some pro-active patients to seek further information on the Internet after discharge, as illustrated in the following quote from one of the pro-active patients. 

“*Well, I care a lot, and I’ll try to follow what I get of medicine, and I can be skeptical if I get another medicine because if I was feeling fine with one, why suddenly switch to another? And yes, it does matter that I get involved in it. If not, I’ll look it up on the Internet myself if I don’t get clear information from the doctor. So, you can do that, and I will do it, or I get help from my grandson because I think they were terrible at informing people about it” (Pt ID #12).*

Again, there was a variation in the pro-active and non-active patients’ approaches when they lacked information about their new medicine, as revealed during the interviews.

#### 3.4.3. Theme 3: No Information on Side Effects

The interviews revealed that almost all interviewed patients would like more discussions about side effects than they have had in the ED. Despite this, several patients explained that no one in the ED enquired about side effects, even though it was a preference of the majority of patients. This is also revealed in the low score in Item 4 “Eliciting patient preferences” of 0.38. However, again, there were differences in how the patients approached this.

The pro-active patients explained that they initiated more discussions on side effects and they were not passively waiting for the healthcare professionals to ask. Almost all patients explained that they found it crucial to discuss side effects, but there was a difference in how active the patients were in securing discussions about it. One of the pro-active patients stated he wanted to hear the physicians’ professional arguments for the given treatment and discuss it. This is exemplified in the following quote from a high-health-literacy patient who was troubled by low blood pressure. The pro-active patient said:

“*You see, I have such a fluctuating blood sugar and blood pressure, and the blood pressure is often too low, and in the last weeks I fell, so I have been extremely anxious about receiving too much medicine. It lowers my blood pressure” (Pt ID #21).*

On the contrary, the more non-active patients acted differently, assuming their physicians had complete control of their medicine since they did not enquire about side effects. This is illustrated in the following quote from a non-active patient.

“*No, that has not been discussed. They haven’t discussed it at all. But I guess they know what’s wrong with me and have looked into it” (Pt ID #5).*

This non-active patient revealed that talk on side effects is something she expected the healthcare professionals to bring up. 

#### 3.4.4. Theme 4: A Preference for Medication Reduction

The interviews revealed that most older patients would prefer less medicine. However, the low score in Item 4 “Eliciting patient preferences” of 0.38 indicates that eliciting patients’ preferences for medicine was not the norm for the healthcare professionals in the ED. Yet again, there seemed to be a distinction between how the pro-active and the non-active patients reacted and whether they brought forward their preferences and viewpoints by themselves or only when asked directly.

Some of the most pro-active patients said that they had tried several times to obtain a medicine reduction, but it had not been easy to reduce the number of medicines. This is illustrated in the following quote from a pro-active patient.


*“I would really prefer to limit my medication intake if possible, and that is something that I have inquired about numerous times. I have just counted how many pills I take daily……and I take 19 medications and 4 vitamins, which is quite a lot, I think” (Pt ID #19).*


Another pro-active patient suggested that the healthcare professionals initiate more discussions and improve dialogue about medicine. This is exemplified in the following quote from another pro-active patient.

“*So, I think it would be nice if they inquired about your medication’s status. But then you have the thought in the back of your head that, uhh, they are so busy, they don’t have time [……].It would be wonderful if they asked if everything is OK with the medication you are receiving. Is there anything else you think should be changed? That would have been fantastic in my opinion” (Pt ID #17)*.

This pro-active patient had preferences for more discussions and an evaluation of her medicine. Still, these preferences had low scores in Item 4, indicating that discussions about “what matters to the patient” rarely happen in the ED unless the patient has high health literacy, is pro-active, and explicitly asks for this kind of informal discussion. The non-active patients, in contrast, did not explicitly state their preferences for medicine reduction because they did not know that deprescribing could be an option; it was more of a desire that became apparent when they were asked directly during the interview.

According to several non-active patients, a short life expectancy and age-related physical impairment, such as a visual impairment, could also reduce the patients’ desire to be involved in medication decisions.

Particularly, one non-active older patient explained that she lacked the desire to participate in medication decision making due to her short life expectancy. Therefore, she had to trust that her medicine would help her. However, when asked directly, this non-active patient also expressed medicine-related ambiguity, stating that she preferred to take less medicine if possible due to side effects, such as dizziness and falls. This is illustrated in the following quote. 

“*You know what, I do not have much time left, so I don’t want to get involved. But I hope that what I get is something that will benefit me in some way. I assume that the physicians have examined everything and that I receive only what I require and nothing else” (Pt ID #2)*.


*Interviewer: “But would you like to take less medicine?”*



*Patient: “Yes, in a way, because there are usually some side effects, and I experienced dizziness and they reduced the diuretics, because I became so dizzy” (Pt ID #2).*


However, knowledge about the patient’s preferences was not revealed, because no one in the medical ED specifically asked for the patients’ medicine preferences.

## 4. Discussion

Using a mixed-methods approach in this study provided an improved understanding of SDM in medication decisions in the ED setting and how it aligns (or differs) with older poly-medicated patients’ preferences and needs.

We discovered that the observed healthcare professionals in the medical ED exhibited a low SDM level during medication-related conversations with little elicitation of patients’ preferences or talks about side effects. A low OPTION score was also found in a study by Olling et al. before implementing SDM in clinical practice, while the OPTION score increased after an SDM patient decision aid was introduced in practice [45]. Patient decision aids help healthcare professionals facilitate the patients’ participation in SDM by making options explicit, providing evidence-based information about the associated benefits and harms. Furthermore, patient decision aids help patients consider what matters most to them about the possible outcomes [46]. Several systematic reviews and meta-analyses revealed that patient decision aids can improve patients’ treatment knowledge and health [47,48]. Further, patient decision aids are highly appreciated by patients as they improve knowledge and awareness of treatment options and support the communication with healthcare professionals, enabling an exchange of information [48]. In a study by Savelberg et al., the authors found that even though the patients in the study felt well informed and were satisfied with the communication, eliciting patient preferences was still rare and the authors recommended that the task of eliciting patients’ preferences very well could be delegated to nurses [49].

The low SDM level observed in our study may be due to differences in the patients’ behavior. We found some active, high-health-literacy patients who secured discussions about different treatment options. In contrast, others were more passive and did not take the initiative to ask questions about their medicine. Weir et al. (2018) also found in a qualitative interview study with older people (75+ years with polypharmacy) that the patients varied considerably in preferences for involvement in decision making about medicine. In contrast to our findings, the authors found a variation in whether the patients were willing to let the healthcare professional deprescribe their medicine or not [21]. In our findings, all patients would prefer to take less medicine, but the non-active patients did not reveal their preferences for less medicine until they were explicitly asked about it, which the healthcare professionals should be aware of. A study by Le Bosquet et al. (2019) also found that some patients prefer not to become involved and highlights that it is the healthcare professionals’ responsibility to assess the amount of involvement the patients want [50]. 

The healthcare professionals should also be aware that the reason patients have a non-active approach to medicine conversations may be explained partly by the patients’ low health literacy, as revealed in a study by Katz et al. (2007), who discovered that persons with inadequate health literacy ask much fewer questions regarding medical issues. Furthermore, patients with low health literacy have less understanding of their medical conditions. Therefore, instead of actively seeking treatment, many of these patients avoid situations where their lack of knowledge is revealed, as they may feel ashamed of their difficulties [51].

Joseph-Williams et al. (2014) investigated patient-reported barriers and facilitators to SDM in a systematic review and found that many patients cannot participate in SDM, rather than they will not participate [52]. This highlights the need for a culture shift in healthcare to manage polypharmacy with a patient-centered SDM approach. Healthcare professionals should pay special attention to the non-active poly-medicated patients’ needs and preferences if the inequality in health should be reduced.

Another systematic review by Eriksen et al. (2020) describes how the unequal power between patients and healthcare professionals may increase the risk that patients will hold back important information about their medication regimens, further complicating treatment. The review highlights how patients’ self-perception was influenced by poly-medication, which was embarrassing for medicine users because of all the associated problems. For example, handling side effects, compatibility problems, barriers to getting prescriptions filled, and cost factors were all problematic. These impediments may affect the patients’ sense of who they are as people [53]. The vulnerability of revealing how difficult it is to manage polypharmacy was also seen in our study, as some of the patients highlighted the importance of “not losing face”. This illustrates how older poly-medicated patients may understate their concern about the medicine if they are not explicitly asked about it, with the risk of them being exposed to adverse drug reactions.

The analysis revealed that many older patients with polypharmacy have preferences for less medicine and more discussions about side effects, because side effects are one of the most important issues according to the patients. However, the overall low OPTION 5 mean score of 8.2 indicates that such talks are rarely happening in the ED. The interviews revealed that the non-active patients rarely addressed their preferences. Jansen et al. (2016) suggest in a study that older people may not always be aware that deprescribing is possible. Hence, healthcare professionals must take the lead to introduce it as an option [15].

Our study found that some non-active patients lacked knowledge about the new medicine they received in the ED due to a lack of information, which could be a potential patient safety threat and result in non-adherence. Furthermore, several patients stated that they still needed more information about why their medicine had to be changed. In addition, a systematic review by Eriksen et al. [53] identified high rates of non-adherence among newly discharged patients with polypharmacy, and studies reveal an increased number of medications prescribed in hospitals is associated with patient non-adherence. This can lead to poor outcomes and patients independently altering their medicine with the risk of adverse events [53] if these issues are not addressed sufficiently in proper medication communication in the ED.

It is healthcare professionals who have the power to reverse the described dynamic, compromising the quality of care (and patient safety). If the healthcare professionals in the ED do not practice SDM and help their patients to trust that it is safe to communicate their concerns and priorities, it may threaten poly-medicated patients’ health and safety. However, the OPTION 5 measurements in our study were low, indicating that there may be potential for improving SDM with a tailored patient decision aid that may help the healthcare professionals in the ED to systematically investigate older poly-medicated patients’ preferences and discuss side effects and harmful medicine. In addition, a tailored patient decision aid may help give information and check the patient’s understanding of the presented options. Yet, SDM may challenge evidence-based medicine because SDM is personalized medicine [16]. It requires a shifting culture if treatment is tailored to the older poly-medicated patients’ overall health, preferences, and life situation and not based strictly on clinical guidelines for each single disease.

### Methodological Strength and Limitations

This study has several strengths and possible limitations, which might have influenced the results. First, the mixed-methods design was a strength as it gave a dual view on SDM in medication decisions in the ED, from both a patient and a healthcare professional point of view. This is essential knowledge as SDM is a social, collaborative process with two persons who interact and influence the conversation. Moreover, our study produced helpful knowledge to guide the design of a “Medication patient decision aid” or a “Conversation guide” in subsequent research, and the study may inspire other researchers in their development of patient decision aids for patients with multimorbidity and polypharmacy. 

Furthermore, it was a strength that we developed a detailed consensus scoring manual supervised by experienced pharmacists to secure consistency in the scores. However, the quality of a mixed-methods study is also dependent on the quality of the research methods used [29]. Concerns may be raised about whether our research was biased because the OPTION 5 measurement was based on field notes and not audio or video recordings of interactions between the patient and healthcare professional. However, the field notes consisted of detailed observations of all medication conversations between the patient and the healthcare professional during the time of observation; so, we would argue that at least the majority of the conversations are documented as they verbally occurred. In addition, the field notes were detailed enough to score with the OPTION 5 instrument. However, no prior sample size calculation was performed for the quantitative measures; so, as such, the estimate should be considered as exploratory. Thus, we cannot reject (or confirm) if the healthcare professionals practiced SDM about medicine with the patient outside the observation period.

To compare our findings with other studies, Couët et al. (2015), in a systematic review of studies using the OPTION instrument, also found low levels of SDM behaviors with an average mean OPTION score of (23 ± 14) on a 0–100 scale in studies where no SDM intervention was used [35]. These scores are still higher compared to our results, which might be due to differences in the patient population, the complexity of the decision in focus, and the characteristics of the acute setting. 

Another potential study limitation is that the patient interviews were over the phone (due to the COVID-19 situation and the patients’ wishes). This resulted in a less “rich” exploration of the patient perspective, allowing the researcher to exert more control, which might also be a limitation to consider. On the other hand, the sample size of 14 interviews in the qualitative part was a strength when compared to other studies. Guest et al. (2006) experimented with data saturation and found that saturation occurred within the first 12 interviews and meta themes occurred as early as six interviews [54]. Furthermore, it is possible that some of the individuals included could have had undiagnosed, mild cognitive impairment, which impacted their ability to be proactive in medical conversations. On the other hand, we do believe it is unlikely that patients had severe cognitive impairment. A Danish validation study of acute medical hospital admissions in the Danish National Patient Registry revealed, among the 127 patients registered with acute admission, 124 were confirmed to be correctly classified according to their diagnosis in the medical records [55].

## 5. Conclusions

In conclusion, our mixed-methods analysis and discussion reveal four overall themes that seem to characterize SDM in medication decisions in the ED setting. We found that many older poly-medicated patients prefer to take less medicine and to have more information and discussions on side effects than they have experienced in the ED. These preferences seem to differ from the healthcare professionals, who observed a low degree of shared decision-making practice in conversations about medicine. However, we also found a variation in preferences for SDM pro-active and non-active patients. The non-active patients prefer to leave medical decision making to the healthcare professionals or are afraid of revealing that they are not in control of their own medicine and, therefore, ask fewer questions, which may be due to their low health literacy. In contrast, pro-active patients take initiative and obtain the information about the medicine that they need. Therefore, healthcare professionals should systematically investigate older poly-medicated patients’ needs to discuss the side effects and the possibility of deprescribing, as these were mostly older patients’ preferences. We would emphasize that the mixed-methods study design provides helpful insight into which specific questions healthcare professionals in the ED could ask older poly-medicated patients regarding their thoughts on their medication to conduct SDM. We will use this information in our following study when tailoring a patient-centered “Guide for medication dialogue” with an SDM approach since no earlier research could advise which specific features SDM concerning polypharmacy in this unique setting with this vulnerable patient group should contain. SDM is vital to securing patient-centered treatment and care and to improving patient experience. However, the consequences of a lack of SDM with older patients with polypharmacy may result in increased inequalities, and the vulnerable patients with low health literacy do not have the possibility of participating in discussions about side effects and treatment. In addition, it might threaten patient safety with the risk of non-adherence and vulnerable older patients may experience unnecessary medication-related admissions and harm, which may be preventable with improved patient-centered medication conversations. 

### Implication for Healthcare Professionals

In the ED, healthcare professionals should be aware that many non-active older patients with polypharmacy ask fewer questions about their medications, despite the fact that they may require knowledge and dialogue the most. The absence of questions may be attributable to a lack of health literacy or a reluctance to reveal that the patients lack medication control.

## Figures and Tables

**Figure 1 ijerph-19-06429-f001:**
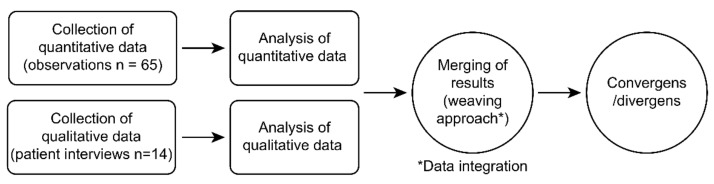
Overview of convergent parallel design.

**Table 1 ijerph-19-06429-t001:** Participant characteristics (patients).

ID	Interview Telephone (T) or Home (H)	Sex (M/F)	Age (years)	Number of Medicines	Home Care or Kindship for Medicine Administration
1	H	M	86	5	No
2	T	F	89	12	Yes
5	T	F	79	12	No
6	T	F	90	7	Yes
7	T	M	81	5	Yes
9	T	F	94	7	No
11	T	M	87	6	No
12	T	F	75	11	Yes
13	T	F	94	6	Yes
15	T	M	78	15	Yes
17	T	F	82	13	No
18	T	M	89	6	No
19	T	F	78	19	No
21	T	M	83	8	Yes

**Table 2 ijerph-19-06429-t002:** Semi-structured interview guide used in the qualitative interviews.

**Theme 1:** Explore whether the patient recalls any medication-related conversations in the ED
1. Opening Question: *Do you recall speaking with anybody (physician, nurse, or pharmacist) about your medication during your ED stay?*
2. *How did you feel about the medication conversations in the ED?*
3. *How did you experience being part of the decision-making process for your medication?*
**Theme 2:** Examine the patient’s experience of being engaged in medication choices in the ED
4. *What are your thoughts on being (or not being) involved, in the way that you did?*
5. *When it comes to your medicine, what is the most crucial thing for you to be engaged in?*
6. *How would you rather make decisions about your medicine?*
**Theme 3:** Examine to see if the patient experienced SDM in conversations with the healthcare professionals about medicine (the Collaborate questionnaire inspired the following three questions)
7. *How did you feel about the healthcare professionals’ efforts to explain your medicine to you?*
8. *How did you feel about the healthcare professionals’ efforts to understand what was most important to you in your medication?*
*9. How was the impression of the effort made to incorporate what matters most to you in your future medication?*

**Table 3 ijerph-19-06429-t003:** Participant characteristics.

Characteristics	Observations Healthcare Professionals, n = 31 (Quantitative)	Interview Patients, n = 14 (Qualitative)
Women, n (%)	18 (58.1)	8 (57.1)
Men, n (%)	13 (41.9)	6 (42.9)
Experiences, mean (years)	14.7	-
Age mean (years)	41.2 (range 26–65)	84.6 (range 75–94)
Number of medications, mean	-	9.4

**Table 4 ijerph-19-06429-t004:** The scores for each OPTION 5 item.

Item #	Behavior	Score					
		0	1	2	3	4	Mean
1. Presenting options	For the health issue being discussed, the clinician draws attention to or confirms that alternate treatment or management options exist or that the need for a decision exists. If the patient rather than the clinician draws attention to the availability of options, the clinician responds by agreeing that the options need deliberation.	41	16.5	4.5	3	0	0.53
2. Establishing a partnership with the patient	The clinician reassures the patient or reaffirms that the clinician will support the patient to become informed or deliberate about the options. If the patient states that they have sought or obtained information prior to the encounter, the clinician supports such a deliberation process.	61	0	1	3	0	0.17
3. Describing pros and cons of options	The clinician gives information or checks understanding about the options that are considered reasonable (this can include taking no action), to support the patient in comparing alternatives. If the patient requests clarification, the clinician supports the process.	51	8	3	2	1	0.37
4. Eliciting patient preferences	The clinician makes an effort to elicit the patient’s preferences in response to the options that have been described. If the patient declares their preference(s), the clinician is supportive.	51	5	5	2	1	0.38
5. Integrating patient preferences into the decision	The clinician makes an effort to integrate the patient’s elicited preferences as decisions are made. If the patient indicates how best to integrate their preferences as decisions are made, the clinician makes an effort to do so.	58.5	1.5	4	1	0	0.19
Total OPTION 5 score: Summed for all 5 items		1.64					
Total OPTION 5 score: Rescaled to 0–100		8.2					

The data shown are the number of observations.

**Table 5 ijerph-19-06429-t005:** The meta-inferences between the two datasets.

Qualitative Findings’ Initial Themes	Meta-Inferences	Quantitative Findings’ OPTION 5 Score and Item Points	Confirmation, Discordance, or Expansion from Findings
Sparse communication about medicine in the ED	*Theme 1: A low degree of SDM in conversations about medicine in the ED* Most patients had not experienced much communication about their medicine and were not even aware that they could have different medicine options.	Total mean score: 8.2 (out of 100)	*Confirmation* Each analysis confirms that there is sparse SDM and communication about medicine in the ED.
Power disparities prevent dialogues about medicine	*Theme 2: A variation in the pro-active and non-active patients’ approach to conversations about medicine* Some pro-active patients were very much involved in decisions about their medicine in contrast to the non-active patients who were divided into two sub-groups. Some patients preferred to leave decisions to the healthcare professionals. The other sub-group was afraid of revealing that they were not in control of their medicine.	Item 2: Establishing a partnership with the patient (Mean points: 0.17) + Item 3: Describing pros and cons of options (Mean points: 0.37)	*Expansion* We found a variation between the pro-active and non-active patients’ approach, which influenced how much information and involvement the patient received and how satisfied they were with the information and involvement.
Talk about side effects	*Theme 3: No information of side effects* Most patients had preferences for more discussions about side effects, but this was rarely discussed or asked for. The low score on Item 4: *Eliciting patient preferences* also reflects that the healthcare professionals rarely investigate patients’ preferences, which also are reflected in theme 4.	Item 4: Eliciting patient preferences (Mean points: 0.38)	*Discordance* There are discordant findings in the two analyses because the Item 4 score is low (Mean 0.38), reflecting that the healthcare professionals rarely investigate patient preferences. However, older poly-medicated patients have preferences for more discussions and information of side effects and the option of reducing medicine, as do the non-active patients, even though they do not ask for it themselves.
Preferences for deprescribing	*Theme 4: A preference for medication reduction* Most patients preferred to take less medicine, but this option was rarely discussed (which also was reflected in the low Item 4 score) unless the pro-active patients asked for it themselves. For non-active patients, they lacked the desire to engage actively in decision making because of vulnerability and limited life horizon, so they had to put trust in their medicine and felt that there was no other choice.		

**Table 6 ijerph-19-06429-t006:** Overall OPTION 5 score.

	Mean	SD	Median	Lower IQR	Upper IQR
Total score	8.2	16.2	0.0	0.0	5.0

## Data Availability

The datasets and analyses used during the study are available from the corresponding author upon request.

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
