# Peer review of "Shared Decision Making with Acutely Hospitalized, Older Poly-Medicated Patients: A Mixed-Methods Study in an Emergency Department"

_ijerph, 2022, doi:10.3390/ijerph19116429_

Round 1

Reviewer 1 Report

I applaud you for pursuing this study regarding shared decision making in the emergency department, with a focus on decisions regarding medication in poly-medicated older adults. This is an important topic given the aging of the population with increasing complexity, number of medications, and likelihood of adverse medications effects leading to morbidity and mortality. I do not find a similar study on literature review.

Your introduction provides a good background to the issues faced by older poly-medicated adults and reasons that SDM regarding medications is important in this cohort. I appreciate the explanation of the study design and found Figure 1 to be particularly helpful in providing a clear explanation of the convergent parallel design which was employed. I found the use of OPTION 5 to be acceptable and appreciate that a conscious effort was made to observe providers for a number of disciplines. It is striking how little SDM was conducted in the ED regarding medication related issues.  Participants who were included in the qualitative data collection appear to be representative of a general sample of older adults, excluding those with cognitive impairment for obvious reasons. I would suggest that some of the individuals include may have had undiagnosed cognitive impairment which impacted their ability to be pro-active in medical conversations.

The results from the quantitative and qualitative portions of the study are clearly presented. Table 5 was particularly useful in presenting the meta-inferences between the data sets. I question if the quotes presented under the four themes would be clearer if presented in a table format for each theme and divided between pro-active and non-active participants. It would also shorten the text of this section which is quite long.

I appreciate the thorough discussion which is presented. I agree that implementing decision aids is critical for improving SDM in the ED and other environments. I would look forward to further study into this area. I applaud the recognition that low health literacy is also an important reason that many older adults do not engage in SDM. It should be screened for and addressed during health encounters.

I would recommend review by a native English speaker to address some inconsistencies and awkwardness of wording and grammar.

Author Response

Response to Editor and Reviewers

Manuscript: ijerph-1730961

”Shared Decision Making with Acutely Hospitalized Older Poly-medicated Patients: A Mixed-method Study in an Emergency Department

Dear

Liz Fang and Reviewer 1:

Thank you very much for your and the reviewers' excellent response to our paper and for providing us with the opportunity to improve it. These are our responses to reviewer one’s comments.

Reviewer 1 comment 1:

I applaud you for pursuing this study regarding shared decision making in the emergency department, with a focus on decisions regarding medication in poly-medicated older adults. This is an important topic given the aging of the population with increasing complexity, number of medications, and likelihood of adverse medications effects leading to morbidity and mortality. I do not find a similar study on literature review.

 Answer:

Thank you very much, we are happy for this comment.

Indeed, as reviewer one we do believe that a focus on decisions regarding medication in poly-medicated older adults in the emergency department, is an important topic given the aging of the population with increasing complexity, number of medications, and likelihood of adverse medications effects. We agree that a similar study is not published

Reviewer 1 comment 2:

Your introduction provides a good background to the issues faced by older poly-medicated adults and reasons that SDM regarding medications is important in this cohort. I appreciate the explanation of the study design and found Figure 1 to be particularly helpful in providing a clear explanation of the convergent parallel design which was employed. I found the use of OPTION 5 to be acceptable and appreciate that a conscious effort was made to observe providers for a number of disciplines. It is striking how little SDM was conducted in the ED regarding medication related issues. 

Answer:

Thank you very much for this feed-back. We appreciate.

 Reviewer 1 comment 3:

Participants who were included in the qualitative data collection appear to be representative of a general sample of older adults, excluding those with cognitive impairment for obvious reasons. I would suggest that some of the individuals include may have had undiagnosed cognitive impairment which impacted their ability to be pro-active in medical conversations.

Answer:

We agree with reviewer one that some of the individuals included could have had undiagnosed mild cognitive impairment, which impacted their ability to be proactive in medical conversations. We have now added the following text in section 4.1.:

“Furthermore, it is possible, that some of the individuals included could have had undiagnosed mild cognitive impairment, which impacted their ability to be proactive in medical conversations. On the other hand, we do believe it is unlikely that patients have had severe cognitive impairment. A Danish validation study of acute medical hospital admissions in the Danish National Patient Registry has revealed among the 127 patients registered with acute admission, 124 were confirmed to be correctly classified according to their diagnosis in the medical records” [55] (page 16, line 640-646)

Reviewer 1 comment 4:

The results from the quantitative and qualitative portions of the study are clearly presented. Table 5 was particularly useful in presenting the meta-inferences between the data sets. I question if the quotes presented under the four themes would be clearer if presented in a table format for each theme and divided between pro-active and non-active participants. It would also shorten the text of this section which is quite long.

Answer:

Thank you very much for this comment. We appreciate and have carefully considered the suggestion, that the quotes in section 3.4 could be presented in table format and divided between pro-active and non-active patients. Our manuscript is heavy on text tables, and we believe section 3.4 present our results in an easier read and more literary approachable format, than a table could.

We have re-read paragraph 3.4 and have added “pro-active” or “non-active” in the text preceding the quotes, thus making sure that all quotes under theme 2, 3 and 4 is attributed to either a pro- or non-active patient”.

 (page 11, line 385, 386, 396, 397, 405), (page 12, line 416, 452, 458),

 (page 13, line 473, 479, 494, 496).

The two quotes under theme 1 are not attributed to pro- or non-active patients, first of all because our findings revealed that there was an overall low degreed of SDM, and secondly as pro- and non-active patients is first introduced under theme 2.    

Reviewer 1 comment 5:

I appreciate the thorough discussion which is presented. I agree that implementing decision aids is critical for improving SDM in the ED and other environments. I would look forward to further study into this area”.

“I applaud the recognition that low health literacy is also an important reason that many older adults do not engage in SDM. It should be screened for and addressed during health encounters.

Answer:

Thank you very much for this comment, and for sharing your point of view.

We are pleased that reviewer one shares our view presented under our discussion, that implementing decision aids is critical for improving SDM in the ED and other environments, and that recognition of low health literacy is an important reason that many older adults do not engage in SDM, and that this factor should therefore also be screened for and addressed during health encounters.

Reviewer one looks forward to further study into this area, and we would therefore like to mention, that we are currently drafting an article under the working title of  “Co-designing a “Guide for Medication Dialogue” to support Shared Decision Making with older poly-medicated patients in the emergency department”, which - as the title implies - could give clinicians a tangible tool to improve SDM

Reviewer 1 comment 6:

I would recommend review by a native English speaker to address some inconsistencies and awkwardness of wording and grammar.

Answer:

Thank you very much for making us aware of gramma errors and awkward use of our English language. Our revised manuscript has now been through the professional editing service that mdpi recommend.

Thank you very much for your useful feed-back, Reviewer 1.

We hope that you will find our revised manuscript suitable for publication in International Journal of Environmental Research and Public Health

Yours Sincerely,

Pia Keinicke Fabricius, PhD student

Clinical Research Center (Section 056)

Amager Hvidovre Hospital, University of Copenhagen

Kettegaard Allé 30

2650 Hvidovre, Denmark

Email: pia.keinicke.fabricius@regionh.dk

Phone +45 51845939

Reviewer 2 Report

Authors have focused their study on provide insight into the extent of shared decision making in medication focused on emergency department. A qualitative and quantitative approaches have been conducted. The scope, development, results and conclusions are linked. The study is focused in a relevant topic.

In my opinion, authors should justify the sample size in both approaches. Second, they should explain in a better way what is different in this study. What new knowledge is provided? Lastly, implications for clinicians should be included.

Author Response

Response to Editor and Reviewers

Manuscript: ijerph-1730961

” Shared Decision Making with Acutely Hospitalized Older Poly-medicated Patients: A Mixed-method Study in an Emergency Department

Dear

Liz Fang and Reviewer 2:

Thank you very much for your and the reviewers' excellent response to our paper and for providing us with the opportunity to improve it. These are our responses to reviewer two's comments.

Reviewer 2 introduction:

Authors have focused their study on provide insight into the extent of shared decision making in medication focused on emergency department. A qualitative and quantitative approaches have been conducted. The scope, development, results, and conclusions are linked. The study is focused in a relevant topic.

Answer: Thank you very much, we are pleased for this comment.

Reviewer 2 comment 1:

In my opinion, authors should justify the sample size in both approaches.

Answer: Thank you very much for making us aware that we have not been clear in our description of our sample sizes. Both approaches are exploratory. We have now added the following text in the paper regarding the quantitative sample size: “However, no prior sample size calculation was performed for the quantitative measures, so as such the estimate should be considered as exploratory”. (page 16, line 622-624).

Regarding the qualitative sample size, we have now added the following text in section 4.1. supporting that data saturation was reached and in support of the sample sizes.

On the other hand, the sample size of 14 interviews in the qualitative part was a strength when compared to other studies. Guest et al. (2006) has experimented with data saturation and found that saturation occurred within the first twelve interviews, and meta themes occurred as early as six interviews” (page 16, line 636-640),                       

Reviewer 2 comment 2:

Second, they should explain in a better way what is different in this study.

What new knowledge is provided?

Answer: Thank you very much for raising these two questions. We have not been able to find a similar study which is published. We are sorry, if we have failed to make it clear what new knowledge is being provided as mentioned, it may reflect the novelty of the subject. However, in section 5.0, we now have inserted the following text:” We would emphasize that the mixed-methods study design provides helpful insight into which specific questions healthcare professionals in the ED could ask older poly-medicated patients regarding their thoughts on their medication to conduct SDM. We will use this information in our following study when tailoring a patient-centered “Guide for medication dialogue” with an SDM approach since no earlier research could advise which specific features SDM concerning polypharmacy in this unique setting with this vulnerable patient group should contain” (page 16, line 663-669).

Reviewer 2 comment 3:

Lastly, implications for clinicians should be included.

Answer: We have now added a new section 5.1. Implication for healthcare professionals with the following text:

“In the ED department, healthcare professionals should be aware that many non-active older patients with polypharmacy ask less questions about their medications, even though they may require knowledge and dialoque, the most. The absence of questions may be attributable to a lack of health literacy or a reluctance to reveal that they lack medication control” (page 17, line 679-684).

Reviewer 2: Thank you very much for this valuable feed-back, which we believe improved our manuscript.

We hope that you will find our revised manuscript suitable for publication.  

Yours Sincerely,

Pia Keinicke Fabricius, PhD student

Clinical Research Center (Section 056)

Amager Hvidovre Hospital, University of Copenhagen

Kettegaard Allé 30

2650 Hvidovre, Denmark

Email: pia.keinicke.fabricius@regionh.dk

Phone +45 51845939